# Estimation of New Regulators of Iron Metabolism in Short-Term Follow-Up After Bariatric Surgery

**DOI:** 10.3390/ijms262110543

**Published:** 2025-10-30

**Authors:** Wojciech Kupczyk, Joanna Boinska, Artur Słomka, Kinga Kupczyk, Marek Jackowski, Ewa Żekanowska

**Affiliations:** 1Department of General, Gastroenterological, and Oncological Surgery, Faculty of Medicine, Ludwik Rydygier Collegium Medicum in Bydgoszcz, Nicolaus Copernicus University in Toruń, 85-067 Bydgoszcz, Poland; kupczykwojciech@cm.umk.pl (W.K.); marek.jackowski@cm.umk.pl (M.J.); 2Pathophysiology Department, Faculty of Pharmacy, Ludwik Rydygier Collegium Medicum in Bydgoszcz, Nicolaus Copernicus University in Toruń, 85-067 Bydgoszcz, Poland; artur.slomka@cm.umk.pl (A.S.); zorba@cm.umk.pl (E.Ż.); 3Department of Hematology and Oncology, National Medical Institute of the Ministry of Interior and Administration, 02-507 Warsaw, Poland

**Keywords:** hepcidin, iron parameters, laparoscopic sleeve gastrectomy

## Abstract

Obesity and bariatric surgery are both associated with disrupted iron homeostasis. These alterations may be mediated by newly identified iron metabolism regulators. The aim of this study was to conduct a short-term, detailed analysis of hepcidin, soluble hemojuvelin, ferroportin, and erythroferrone—as well as whole-body composition—before and five months after sleeve gastrectomy. This approach may help elucidate the potential impact of bariatric surgery on iron metabolism and the timing of these changes. The study included 40 obese patients aged 26–64 eligible for laparoscopic sleeve gastrectomy. Iron parameters were assessed with immunoenzymatic methods. We found significantly increased iron levels (79 µg/dL vs. 95 µg/dL, *p* = 0.0016) as well as reduced hepcidin concentrations five months after bariatric surgery (54.46 ng/mL vs. 33.88 ng/mL, *p* = 0.0177). The change in the reduction in mean body fat (delta MBF) and body fat percentage (delta BPF) was positively associated with delta hepcidin levels with correlation coefficients of R = 0.36 (*p* = 0.0228) for MBF and R = 0.42 (*p* = 0.0070) for BPF. Moreover, significant correlations were observed between the reduction in body fat and soluble hemojuvelin (R = 0.31 *p* = 0.0489 for MBF) (R = 0.45 *p* = 0.0032 for PBF). No patient showed laboratory signs of iron deficiency. Decreased serum hepcidin levels observed five months after sleeve gastrectomy are associated with improved iron status, as indicated by increased serum iron and red blood cell indices. Positive correlations between body fat reduction and both hepcidin and soluble hemojuvelin levels suggest that the resolution of adipose tissue-related inflammation may contribute to improved iron bioavailability.

## 1. Introduction

Obesity is a complex metabolic condition that can markedly affect iron metabolism, and the role of bariatric surgery in addressing these alterations is clinically important. The underlying mechanisms behind iron dysregulation associated with obesity and its treatment are diverse and interconnected, with hepcidin emerging as a key regulator in this complex interplay [1]. Obesity leads to an excess of adipose tissue, which triggers a state of chronic low-grade inflammation. In response to the resulting hypoxia, adipokines such as leptin, visfatin, omentin, resistin, and tumor necrosis factor-alpha (TNF-α) are produced particularly by visceral white adipose tissue (vWAT) [2]. Furthermore, the accumulation of adipose tissue leads to immune cell infiltration, especially by macrophages, which produce interleukin-1 (IL-1) and interleukin-6 (IL-6), cytokines that promote both local and systemic inflammation [3]. Moreover, macrophages, as cells of the inflammatory process, are an important effector of hepcidin action, decreasing the release of iron from these cells into the bloodstream [4].

This proinflammatory environment in obesity stimulates the synthesis of acute phase protein-hepcidin, a key regulator of iron homeostasis, leading to obesity-induced disturbances in iron metabolism [5,6]. Although hepatic hepcidin production is predominant, some experimental studies have shown that adipocytes can also produce hepcidin, but at a significantly lower level than the liver [7]. Elevated hepcidin levels downregulate ferroportin, the major cellular iron exporter, which in turn restricts iron availability and promotes iron deficiency, especially when combined with dietary habits that limit iron intake [8]. There are several pathways involved in hepcidin gene regulation. In addition to inflammation, hemojuvelin and erythroferrone also play significant roles in this process. The activity of hemojuvelin depends on its form—membrane-bound or soluble. The membrane-bound form, as part of the major iron-sensing receptor complex bone morphogenetic proteins (BMP), promotes the upregulation of hepcidin. In contrast, the soluble form competes with BMPs for receptor binding, thereby inhibiting hepcidin expression. It is a complex negative feedback system functioning as a hepcidin-hemojuvelin-ferroportin axis. Erythroferrone, a hormone produced by erythroblasts in response to increased erythropoietin levels, suppresses hepcidin gene expression, enhancing iron bioavailability for erythropoiesis [8,9]. Altered iron metabolism in obesity contributes to hyperferritinemia and may exacerbate liver injury and dysfunction in non-alcoholic fatty liver disease (NAFLD). However, ferritin levels also rise as part of the inflammatory response, so an elevated ferritin concentration does not necessarily indicate absolute iron overload [10]. In obesity-related chronic inflammation, increased ferritin reflects its role as an acute-phase protein and may modulate iron distribution and availability, potentially influencing serum iron levels and overall iron homeostasis.

The treatment of morbid obesity also significantly affects iron metabolism. Laparoscopic sleeve gastrectomy (LSG) is one of the two main types of bariatric surgery, along with Roux-en-Y gastric bypass (RYGB), and both show similar short- and medium-term outcomes in terms of weight loss and a low rate of perioperative complications. Recently, LSG has become the most widely performed bariatric procedure and is often regarded as the first-line surgical option [11,12,13]. In the long-term follow-up of patients after LSG, satisfactory results are observed in mean percentage excess weight loss (%EWL) and in the remission of type 2 diabetes, dyslipidemia, and hypertension [14]. However, the excess body weight achieved after surgery shows considerable fluctuations over time in a substantial proportion of patients [15].

Weight reduction—the primary goal of bariatric surgery—lowers systemic inflammation and, via hepcidin, may enhance iron bioavailability in circulation. However, from another perspective, removing a substantial portion of the stomach can impair the conversion of dietary iron to its absorbable ferrous (Fe^2+^) form due to hypochlorhydria. As a result, less iron is available for absorption in the duodenum [16]. Additionally, adherence to postoperative recommendations, poor dietary habits, and socioeconomic status may influence the long-term maintenance of iron homeostasis, as well as the balance of other micro- and macronutrients. Many studies indicate that despite the use of various forms of perioperative iron supplementation and prophylaxis against iron deficiency after surgery, a significant percentage of patients still develop anemia, and according to some sources, this condition tends to occur with a delay. Iron deficiency following LSG affects 10 to 30% of patients, with an increasing tendency beginning in the first year after bariatric surgery [17,18,19,20].

While changes in standard iron parameters following bariatric surgery have been relatively well described, there is still a lack of studies exploring newly identified iron regulatory markers, particularly with body composition. Therefore, the present study aimed to assess changes in the concentrations of hepcidin, hemojuvelin, ferroportin and erythroferrone before and 5 months after surgery, and to analyze the relationships between these parameters and fat reduction.

## 2. Results

The study group consisted of 40 participants who underwent LSG, including 28 women and 12 men. The age range of the participants was 26 to 64 years (median: 41 years), and the median BMI was 38.8 kg/m^2^ (IQR: 36.1–44.1) (Table 1). Many of the obese patients had comorbidities, the most common being hypertension, hypothyroidism, and diabetes. Post-LSG patients experienced a statistically significant reduction in BMI, which correlated with improvements in most assessed body parameters, including MBF and percent body fat (PBF), and the mean percentage of excess weight loss (% EWL) reached 52%, calculated based on the amount of weight lost relative to each patient’s excess body weight. Furthermore, within five months following surgery, improvements were observed in most biochemical parameters of liver and kidney function; however, their concentrations remained within the reference range both before and after LSG.

As presented in Table 2, we found a significant increase in iron concentration (*p* = 0.0016), as well as a reduction in hepcidin levels (*p* = 0.0177), five months after LSG compared to preoperative values. Individual patient results for these two parameters before and after surgery are shown in Figure 1.

Improved iron availability was reflected in red blood cell indices such as MCV, MCH, MCHC, and an increase in RDW-SD. However, no changes in hemoglobin levels were observed. Furthermore, no significant changes in erythroferrone, soluble hemojuvelin, ferroportin, and transferrin levels were observed over 5 months post LSG.

Based on consecutive measurements of iron parameters and body composition indices (before and after LSG), changes over time (deltas) were calculated as the difference between follow-up and baseline values. A reduction in body fat was significantly associated with changes in iron regulators. Specifically, changes in mean body fat (ΔMBF) and body fat percentage (ΔPBF) showed positive correlations with changes in hepcidin levels (R = 0.36, *p* = 0.0228 and R = 0.42, *p* = 0.0070, respectively). Similarly, significant positive correlations were observed between both ΔMBF and ΔPBF and changes in soluble hemojuvelin levels. In contrast, a significant negative correlation was found between ΔPBF and changes in transferrin levels (Table 3).

## 3. Discussion

Our aim was to investigate iron homeostasis in the context of significant postoperative weight loss and body composition indices. To achieve this, we focused on novel regulatory parameters such as hepcidin, soluble hemojuvelin, ferroportin, and erythroferron during the period when postoperative weight loss becomes significant.

Our current study revealed that LSG leads to significant changes in iron metabolism in the early postoperative period including decreased hepcidin levels and increased serum iron levels, with no laboratory evidence of iron deficiency in any patient. Furthermore, the positive correlations between changes in both PBF and MBF with hepcidin levels support the association between obesity-related inflammation and hepcidin as an acute-phase reactant. In this context, substantial weight reduction appears to lower hepcidin levels. This, in turn, reduces the internalization and degradation of ferroportin, enhancing iron export from enterocytes and macrophages into the circulation. Based on the known physiological mechanisms of iron regulation, the less hepcidin the more iron is available to the variety of iron-dependent processes such as hemoglobin synthesis. Additionally, significant correlations between reductions in fat mass (both absolute and percentage) and soluble hemojuvelin levels may reflect a shift toward the restoration of iron homeostasis and a reduced need for hepcidin suppression, which was significantly lower five months after SG. Although the inhibition of hepcidin activity can be influenced by erythroferron and transferrin, no significant pre- to postoperative changes in these parameters were observed 5 months after surgery. Considering the slight improvements in red blood cell indices and liver enzyme levels (ALT, AST), these findings indicate that iron balance was effectively maintained during the first five months following LSG, reflecting beneficial effects on both iron metabolism and liver function. Taken together, these results suggest that the reduction in inflammation plays a central role in maintaining iron levels above the norm in this study group.

The results of our study correspond to the findings from other research that highlight the beneficial impact of bariatric surgeries on iron status during the first six months after surgery. Studies by Tussing-Humphreys et al. (2010) and Cepeda-Lopez et al. (2016) [21,22] revealed a significant decrease in IL-6 and hepcidin levels six months after bariatric surgery. Additionally, elevated hemoglobin levels and increased iron isotope absorption were observed. On the other hand, Marin et al. (2017) [23] found that the reduction in BMI, C-reactive protein, and ferritin levels may not lead to a magnitude decrease in hepcidin levels six months after RYGB. However, micronutrient supplementation before and after surgery, when compared to standard postoperative care, positively influenced iron status and total iron-binding capacity. The fact that the type of bariatric surgery differs among cited studies may significantly impact the results. Nevertheless, there is no clear consensus on whether any of these procedures require more advanced supplementation of protein, or macronutrients. Misra S. et al. (2020) and Wiese ML et al. (2023) concluded that both RYGB and LSG require similar dietary management and are associated with comparable rates of micronutrient deficiencies [24,25].

An interesting study by Pihan-Le Bars F. et al. (2016) [26] found a direct connection between adipose tissue and hepcidin gene expression. They found that *HAMP* was highly expressed in visceral adipose tissue of obese individuals, while adiponectin mRNA and transferrin receptor expression were decreased. Moreover, they observed a significant correlation between hepcicidin levels and adiponectin, highlighting the link between obesity, fat tissue and major iron regulator. The results of our study seem to align with this aspect. Significant correlations between the reduction in adipose tissue indices and hepcidin levels may reflect a regulatory mechanism of hepcidin gene expression.

In our study, ferroportin levels were low, with no significant differences observed between pre- and postoperative measurements. There are no studies in the available literature assessing the concentration of this protein after bariatric treatment. Our results partially correspond to the findings reported by Gajewska et al. (2018) [27], who observed significantly lower ferroportin concentrations in the blood of children with obesity compared to normal-weight children. It is worth noting that the degree of obesity in the cited study was relatively mild, with BMI values ranging from 19.2 to 30.3. In contrast, our study included patients with morbid obesity, which may account for the markedly low ferroportin levels. At this point, the question arises as to the origin of “circulating” ferroportin, which is typically described as a transmembrane protein expressed in enterocytes and macrophages [28]. It should be noted that current assays do not distinguish between these sources, which represents a methodological limitation. Whether its presence in the circulation results from enterocyte membrane shedding or the release of extracellular vesicles remains to be elucidated. Therefore, the clinical significance of circulating ferroportin changes should be interpreted cautiously.

We did not observe significant changes in soluble hemojuvelin levels when comparing preoperative and postoperative values. However, there was a noticeable trend toward lower levels five months after LSG. Additionally, we found significant positive correlations between fat loss and changes in soluble hemojuvelin levels. This appears to be the first report addressing soluble hemojuvelin dynamics in the context of bariatric surgery. To date, this topic has not been explored. A study by Luciani et al. (2011) reported significantly elevated soluble hemojuvelin levels in obese females selected for bariatric surgery; however, the authors did not assess postoperative values of iron regulatory markers [29].

This study has several limitations. It was conducted at a single center, and the sample size was relatively small; thus, subgroup analyses (e.g., by gender or comorbidities) may have compromised statistical validity. Furthermore, the analysis focused only on the first five months after surgery; it would be valuable to include intermediate time points, e.g., at 1 and 3 months postoperatively. Some variables remain difficult to control, such as patients’ physical activity, which may differ from self-reported levels, as well as medication changes post-surgery and menstrual status in women. Although patient compliance with lifestyle modifications and postoperative nutrition was supervised by a professional dietitian, it remains subjective.

Our findings highlight the relationship between iron metabolism and fat reduction following LSG, suggesting that restrictive bariatric surgery not only reduces body weight but also positively influences iron homeostasis at 5 months postoperatively. These results can also be considered from a broader perspective, as iron imbalance may affect the central nervous system and cognitive function [30,31]. Given the chronic stress often experienced by bariatric patients due to long-standing obesity, metabolic improvements are closely linked to overall well-being. Our study supports a better understanding of early changes in iron metabolism after bariatric surgery and provides a basis for further research to explore these mechanisms in larger cohorts and over longer follow-up.

## 4. Material and Methods

### 4.1. Study Population

The study group consisted of 40 successive obese patients eligible for bariatric LSG, treated at the Department of General, Gastroenterological, and Oncological Surgery, Collegium Medicum, Nicolaus Copernicus University (Toruń, Poland) between 2022 and 2023 (Figure 2). Patients were qualified for surgical treatment based on the European Association for Endoscopic Surgery guidelines for bariatric surgery and the Polish guidelines from 2020 [32,33].

All patients in the study were Caucasian and had a long history of obesity, having previously undertaken multiple unsuccessful attempts at weight loss. Patients underwent a comprehensive evaluation, including assessments by a bariatric surgeon, anesthesiologist, dietitian, psychologist or psychiatrist, cardiologist, pulmonologist, and other specialists as indicated, and received preoperative dietitian support aimed at achieving a 5–10% reduction in body weight and optimizing body composition before surgery, in accordance with the International Federation for the Surgery of Obesity and Metabolic Disorders and the American Society for Metabolic and Bariatric Surgery (IFSO/ASMBS) recommendations. Comprehensive preoperative assessments included laboratory tests, nutritional and organ function evaluations, upper gastrointestinal endoscopy, imaging studies, spirometry, and echocardiography, with additional tests performed as clinically indicated. Perioperative care was conducted according to Enhanced Recovery After Bariatric Surgery (ERABS) protocols, which are designed to optimize patient outcomes and minimize complications.

The inclusion criterion was obesity, defined by a body mass index (BMI) of 40 kg/m^2^ or greater, or a BMI in the range of 35–40 kg/m^2^ with comorbidities who were expected to benefit from surgically induced weight loss. Subjects were excluded from the study if they had a history of previous bariatric surgery or met any of the following exclusion criteria: severe medical conditions (heart or lung disease), active oncological treatment, substance abuse, significant mental health issues, a history of anemia or iron metabolism disorders, use of iron supplementation other than a multivitamin preparation containing iron, or diabetes mellitus lasting more than 5 years. The pre-, peri-, and postoperative periods included full dietary support at every stage of treatment and long-term follow-up care. The preoperative diet was individually tailored for a minimum of three months, providing 1.2–1.5 g of protein per kg of ideal body weight, 10–18 mg/day of iron (with at least 50% from heme sources), and a minimum of 100 mg/day of vitamin C with non-heme iron-containing meals. The only routine preoperative supplementation was vitamin D at 4000 IU/day.

Following surgery, due to reduced dietary intake and limited food volume, routine supplementation was initiated from the second postoperative week in the form of a daily oral capsule containing a multivitamin complex with 28 mg of iron, 100 mg of vitamin C, B-group vitamins, vitamin D, and additional minerals to meet recommended requirements. Protein intake was maintained at 1.2–1.5 g/kg of ideal body weight per day, including protein provided by commercial supplements, in accordance with clinical protocol. Additionally, all patients received a proton pump inhibitor once daily.

Blood samples for all laboratory measurements of iron status and biochemical markers were drawn a day before surgery and five months after, during the standard follow-up visit. Anthropometric measurements were also assessed this same selected time points (before and after surgery). All study participants provided informed written consent. The study was approved by the Bioethics Committee of the Nicolaus Copernicus University in Toruń (KB 82/2021; approval date: 19 January 2021).

### 4.2. Anthropometric and Body Composition Measurements

All patients were evaluated in terms of body weight and height. Weight was measured using a digital scale (±0.1 kg), and height was measured with a stadiometer (±1 cm). These basic values were used to calculate BMI according to the standard formula. Moreover, detailed body composition, including mass of body fat (MBF), percent body fat (PBF), extracellular water (EW), intracellular water (ICW), and excess weight (EW), was assessed by body composition analyzer (Jawon Medical X-contact 350, Jawon, Seongnam, Republic of Korea).

### 4.3. Laboratory Analysis

Blood samples were obtained at two time points: at baseline and during the follow-up examination. Blood was drawn into two plastic tubes containing K2EDTA and a clot activator (Vacutainer System, BD Biosciences, Franklin Lakes, NJ, USA). After centrifugation of the samples (at 2500× *g* at +4 °C for 20 min), plasma and serum were stored at −80 °C until further use in assays.

Complete blood count was analyzed in whole blood using automated hematology analyzer (Sysmex XN-3000) (Kobe, Japan). Serum iron, folic acid, and vitamin B12 concentrations, as well as biochemical parameters such as creatinine, alanine aminotransferase, aspartate aminotransferase, urea, and creatinine, were analyzed at the central hospital laboratory.

The serum hepcidin levels were measured by Intrinsic Hepcidin IDx ELISA (ICE-007, IntrinsicDx, La Jolla, CA, USA). Hemojuvelin (HJV) was quantified by an ELISA assay provided by Cloud-Clone Corp. (SEB995Hu, Katy, TX, USA). Serum transferrin concentration was determined using highly sensitive two-site enzyme linked immunoassay test (Catalog No. IHUTFKT) (Innovative Research, Inc., Innovative Research, Inc., Novi, MI, USA). Ferroportin (FPN) levels were measured using ELISA technique (SEC489Hu, Cloud-Clone Corp., Katy, TX, USA). Erythroferrone (ERFE) concentrations were determined using Intrinsic Erythroferrone IE^TM^ ELISA (SKU# ERF-001) (IntrinsicDx, La Jolla, CA, USA).

All laboratory measurements were performed according to the producer’s recommendations and in accordance with good laboratory practice.

### 4.4. Statistical Analysis

All statistical calculations were conducted using Statistica 13.3 (TIBCO Software Inc., Palo Alto, CA, USA). The distribution of data was verified with the Shapiro–Wilk test. Continuous variables are presented as medians with interquartile ranges (IQR), while categorical variables are shown as numbers and percentages. Comparisons between groups and over time for non-normally distributed data were performed using the Mann–Whitney U test and the Wilcoxon signed-rank test, respectively. Differences in categorical variables were analyzed with the Chi-squared test. Associations between variables were evaluated using Spearman’s rank correlation coefficient. Statistical significance was established at a two-tailed *p*-value < 0.05.

## 5. Conclusions

Decreased serum hepcidin levels observed five months after SG are associated with improved iron status, as indicated by increased serum iron and red blood cell indices. Positive correlations between body fat reduction and both hepcidin and soluble hemojuvelin levels suggest that the resolution of adipose tissue-related inflammation may contribute to improved iron bioavailability.

## Figures and Tables

**Figure 1 ijms-26-10543-f001:**
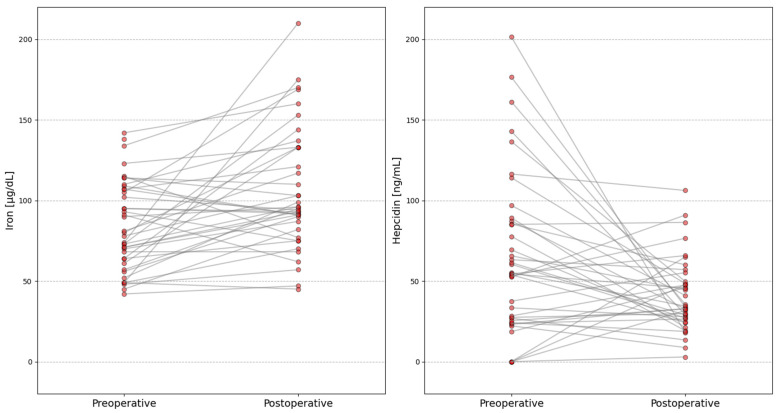
Changes in Iron and Hepcidin levels before and after laparoscopy sleeve gastrectomy (LSG). Red dots and lines indicate individual patient trajectories from pre- to postoperative measurements.

**Figure 2 ijms-26-10543-f002:**
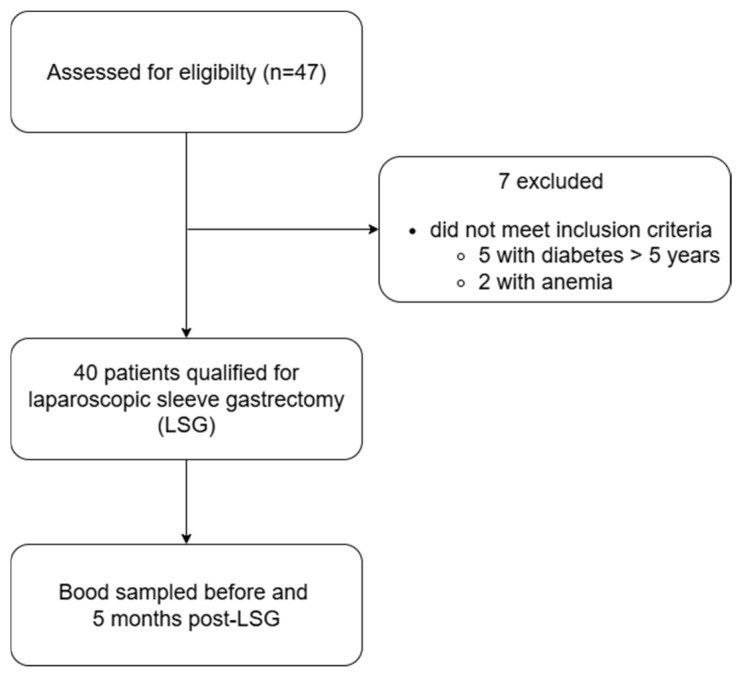
Flow diagram of patient selection.

**Table 1 ijms-26-10543-t001:** Basic characteristics of the study groups in terms of gender, age, body weight and composition, body mass index (BMI), selected biochemical markers, and comorbidities.

Variable	Preoperative	5 Months Postoperative	Z	*p*-Value	r
Age, y	41 (33–45)	-		-	
Male-sex, n (%)	12 (30)	-		-	
Height, m	1.71 (1.63–1.79)	-		-	
Weight, kg	114 (103–134)	88 (78–109)	5.44	<0.0001	0.87
BMI, kg/m^2^	38.8 (36.1–44.1)	29.8 (27.6–35.7)	5.44	<0.0001	0.87
MBF, kg	46.50 (41.40–52.70)	29.00 (25.00–37.40)	5.44	<0.0001	0.87
PBF, %	42.60 (34.80–44.70)	33.60 (28.80–38.20)	5.44	<0.0001	0.87
ECW, kg	19.00 (17.20–24.60)	16.35 (15.10–22.40)	5.23	<0.0001	0.85
ICW, kg	27.40 (25.40–35.90)	23.75 (21.80–32.30)	5.32	<0.0001	0.86
EW, kg	48.30 (39.40–64.90)	22.05 (16.30–38.30)	5.37	<0.0001	0.87
Biochemical markers					
ALT, U/L	28.50 (21.50–41.0)	19.50 (13.00–24.00)	4.89	<0.0001	0.77
AST, U/L	24.00 (21.00–29.50)	20.50 (19.00–26.00)	2.85	0.0044	0.48
Urea, mg/dL	27.55 (24.30–32.80)	24.80 (21.70–29.70)	2.38	0.0174	0.38
Creatinine, mg/dL	0.72 (0.65–0.85)	0.71 (0.67–0.83)	0.17	0.8618	0.03
Folic acid, ng/mL	7.50 (5.40–10.20)	7.85 (5.45–13.45)	2.06	0.0391	0.33
Vitamin B12, pg/mL	363.00 (266.5–433.0)	339.50 (291.0–401.0)	0.58	0.5625	0.09
Comorbidities					
Hypertension, n (%)	20 (50)	-		-	
Diabetes mellitus, n (%)	6 (15)	-		-	
Hypothyroidism, n (%)	10 (25)	-		-	
PCOS, n (%)	2 (0.5)	-		-	
Dyslipidemia, n (%)	2 (0.5)	-		-	

Data are presented as median (IQR) for continuous variables, and n (%) for categorical variables; Abbreviations: ALT, alanine aminotransferase; AST, aspartate aminotransferase; BMI, body mass index; ECW, extracellular water; EW, excess weight; ICW, intercellular water; MBF, mass of body fat; PBF, percent body fat; PCOS, polycystic ovary syndrome.

**Table 2 ijms-26-10543-t002:** Pre- and postoperative complete blood count and iron regulatory markers in bariatric surgery patients.

Variable	Preoperative	5 Months Postoperative	Z	*p*-Value	r
Complete blood count					
RBC, 10^12^/L	4.89 (4.52–5.13)	4.81 (4.41–5.03)	1.36	0.1736	0.22
HGB, g/dL	14.15 (13.25–15.10)	14.00 (13.05–15.00)	1.11	0.2672	0.18
HCT, %	40.95 (38.90–43.70)	41.30 (38.85–43.75)	0.41	0.6794	0.07
MCV, fL	83.80 (82.55–86.20)	86.35 (84.45–87.85)	4.33	0.0002	0.69
MCH, pg	29.20 (28.25–29.75)	29.60 (28.70–30.10)	2.64	0.0084	0.42
MCHC, g/dL	34.30 (33.80–35.00)	34.20 (33.75–34.60)	3.20	0.0014	0.51
RDW-SD fL	40.15 (38.90–42.25)	41.40 (40.20–43.85)	3.90	0.0001	0.62
Iron regulatory parameters					
Iron, µg/dL	79.00 (62.50–107.00)	95.00 (82.00–133.00)	3.15	0.0016	0.51
Erythroferron, ng/mL	0.99 (0.60–1.72)	1.06 (0.65–2.39)	0.96	0.3385	0.16
Hepcidin, ng/mL	54.46 (26.18–86.36)	33.88 (26.44–49.72)	2.37	0.0177	0.38
Soluble hemojuvelin, ng/mL	15.10 (8.31–18.37)	11.64 (6.57–15.01)	0.85	0.3971	0.13
Ferroportin, ng/mL	0.01 (0.01–0.03)	0.02 (0.02–0.03)	0.32	0.7457	0.06
Transferrin, ng/mL	940.91 (534.18–1389.61)	1073.14 (809.43–1539.54)	0.83	0.4046	0.13

Data are presented as median (IQR) for continuous variables. Abbreviations: RBC, red blood cells; HGB, hemoglobin; HCT, hematocrit; MCV, mean corpuscular volume; MCH, mean corpuscular hemoglobin; MCHC, mean corpuscular hemoglobin concentration; RDW-SD, red cell distribution width—standard deviation.

**Table 3 ijms-26-10543-t003:** Analysis of significant correlations between changes in body composition and changes in iron metabolism parameters.

	R	*p*
ΔMBF and ΔHepcidin	0.36	0.0228
ΔPBF and ΔHepcidin	0.42	0.0070
ΔMBF and ΔSoluble hemojuvelin	0.31	0.0489
ΔPBF and ΔSoluble hemojuvelin	0.45	0.0032
ΔPBF and ΔTransferrin	−0.37	0.0192

Abbreviations: MBF, mass of body fat; PBF, percent body fat; delta MBF (ΔMBF)  =  follow-up MBF−baseline MBF. Similarly, ΔPBF, ΔHepcidin, ΔSoluble hemojuvelin, ΔTransferrin were calculated as the values at follow-up minus the values at baseline.

## Data Availability

The data presented in this study are available upon request from the corresponding author.

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
