# Peer review of "Estimation of New Regulators of Iron Metabolism in Short-Term Follow-Up After Bariatric Surgery"

_ijms, 2025, doi:10.3390/ijms262110543_

Round 1

Reviewer 1 Report

Comments and Suggestions for Authors

The study addresses an interesting question but has significant data gaps and methodological limitations that need to be clarified. There are  some missing data that need to be completed:

1- As regards patient demographics and selection: No information on ethnicity/race of participants, missing data on previous weight loss attempts, no details on duration of obesity, no reports of surgical complications or adverse events and lastly no mention of patient dropout or loss to follow-up.

2-No clear statement on whether patients received iron supplementation during the 5-month period without dosage, formulation, and compliance with supplementation protocols with no mention of pre-operative iron supplementation practices.

3- As regards diet, the baseline dietary iron intake not quantified with missing post-operative dietary adherence metrics and neither reported protein intake levels (crucial for iron absorption) nor vitamin C intake (affects iron bioavailability).

4- As regards laboratory reference ranges, no mention of the normal ranges for hepcidin, hemojuvelin, ferroportin, erythroferrone) that makes interpretation of “normal” vs. “abnormal” values impossible.

5- Concerning statistical Details, there are no reports of effect sizes; confidence intervals; and power calculation. 

6- Authors discussed inflammation extensively), however, they did not measure IL-6, CRP, TNF-α levels. This is a major gap given the inflammatory hypothesis central to conclusions.

7- Data regards time course are scarce (only two time points; baseline and 5 months) without intermediate measurements to track trajectory of changes

8- The ferritin Levels are mentioned in discussion but not reported in results tables.

9- The following confounding variables are not controlled (medication changes post-surgery, assessment of Physical activity levels, and consideration of menstrual status in women)

10- I need an explanation of the folowing selection criteria (exclusion of patients with diabetes >5 years; patients taking iron supplements) limiting generalizability

11- I will recommend authors to include the following if feasible: ferritin measurements in results section, inflammatory markers (CRP, IL-6) to support mechanistic claims, documentation of iron supplementation protocols, tracking adverse events systematically, adding intermediate time points (e.g., 1, 3 months post-surgery) and assessing dietary iron intake using validated food frequency questionnaires.

12-I would recommend authors to correct transferrin units -verify if these should be mg/dl, to provide reference ranges for all novel markers in Table 2, to add individual patient trajectories for key markers (not just pre-post scatter)

13- I recommend inclusion of flow diagram showing patient selection and any dropouts

14- As regards the discussion section, I recommend addressing the discrepancy with literature on iron deficiency rates, emphasing clearly the limitations of  ferroportin measurement. On the other hand, I suggest discussing the clinical significance of findings, not just statistical significance with explaination why no inflammatory markers were measured despite central hypothesis.

15- Add a limitation section that addresses the temporal limitations of 5-month follow-up more critically.

Reviewer 2 Report

Comments and Suggestions for Authors

Thank you for the opportunity to review this study. The authors have conducted a prospective study evaluating iron homeostasis after sleeve gastrectomy, at 5 months postoperatively. Below are my comments to the authors:

- It is well-known that metabolic bariatric surgery might lead to iron deficiency particularly after RYGB. The authors here found that iron levels were increased after SG at 5 months, but in the normal range. I think that this shows an improved availability of iron rather than "increased absorption". Because someone might interpret that SG leads to increased iron levels (which is contradictory to the current literature and guidelines), you should better clarify that "improved iron bioavailability is observed".

- In the Introduction, regarding mechanisms, mention also the changes in ferritin levels in the inflammatory obese state and whether this affects iron levels.

- In the methods, what were the exact indications for performing SG and not RYGB? Was it patient preference, what were the exact selection criteria? Clarify this.

- Clarify also in methods the following. Did the patients take any supplements after the operation? Postoperatively did the patients take any PPIs?

- In this study you assessed outcomes in five months after SG. Regarding the long-term, SG is related with weight regain and this could further affect micro-nutrient deficiencies as well. A recent study showed that different clinical response is observed after SG in terms of weight loss and this could affect micro-nutrient status as well. As a future perspective it would be interesting to assess these findings in the long-term and in patients with weight recurrence. You could add this study in the references. Lampropoulos, Charalampos et al. “Critical Time Points for Assessing Long-Term Clinical Response After Sleeve Gastrectomy-A Retrospective Study of Patients with 13-Year Follow-Up.” Obesity surgery vol. 35,2 (2025): 571-581. doi:10.1007/s11695-024-07659-7

Overall, the methods are clearly reported, enabling reproducibility and all the appropriate limitations are acknowledged.

Round 2

Reviewer 1 Report

Comments and Suggestions for Authors

The authors responded positively to my comments